ARTICLES

# Large-scale microbiome data integration enables robust biomarker identification

Liwen Xiao[1,2,5], Fengyi Zhang[1,5] and Fangqing Zhao [1,2,3,4] ✉

The close association between gut microbiota dysbiosis and human diseases is being increasingly recognized. However, contradictory results are frequently reported, as confounding effects exist. The lack of unbiased data integration methods is also impeding the discovery of disease-associated microbial biomarkers from different cohorts. Here we propose an algorithm, NetMoss, for assessing shifts of microbial network modules to identify robust biomarkers associated with various diseases. Compared to previous approaches, the NetMoss method shows better performance in removing batch effects. Through comprehensive evaluations on both simulated and real datasets, we demonstrate that NetMoss has great advantages in the identification of disease-related biomarkers. Based on analysis of pandisease microbiota studies, there is a high prevalence of multidisease-related bacteria in global populations. We believe that large-scale data integration will help in understanding the role of the microbiome from a more comprehensive perspective and that accurate biomarker identification will greatly promote microbiome-based medical diagnosis.

Microbiomes in the human body have profound impacts on many aspects of human health, especially those in the gut, which are closely associated with the occurrence and development of many diseases[1,2]. Comprehensive evaluations of the relationship between microbiota and disease are significant to improving health. Biomarkers correspond to biological indicators to measure and evaluate the biological states of individuals, such as differentially expressed genes or differentially abundant bacteria[3]. Accurate identification of biomarkers helps to facilitate clinical diagnosis and improves clinical prognosis prediction[4,5]. Most previous studies have identified key bacteria as biomarkers based on variation in abundance between healthy and diseased groups[6–8]. However, confounding factors between studies often mask the real features of microbial communities and thus may lead to unreliable conclusions. Although several studies have sought to address the challenge by correcting statistical parameters or microbial profiles[9,10], the dependence on additional clinical information limits their applications. Statistical tools such as combat[11] and limma[12], which were developed to remove batch effects in the analysis of microarray expression data, also exhibit poor performance due to the sparsity feature of the microbial datasets. Consequently, computational methods for the integration of microbiome data from different cohorts are urgently needed.

In the human gut, the interaction of microbial species, rather than microbes alone, maintains community structure and provides a stable environment for commensals, in which co-occurrence networks contribute to an understanding of the relationship between different taxa[13–15]. A number of studies have demonstrated that the application of co-occurrence networks can simplify the identification of disease-related biomarkers and thus improve clinical prediction models[16–18]. Nevertheless, great challenges in network-based microbiome analysis remain, especially when integrating networks of multiple cohorts. For example, different cutoff selections substantially alter the topological structures of networks and thus

correspond to different microbial interactions. In addition, the sample size in microbiome studies also influences the network structure. The most common tactic in data integration is to combine networks directly based on microbial interaction pairs. However, this kind of method fails to consider divergence among different datasets. An integrated network ranking approach to predict regulatory genes involved in the host response has been proposed[19]. Taking perturbation into account, its application in microbial analysis is still limited.

In this study we have developed an algorithm called Network Module Structure Shift (NetMoss), which focuses on the shift of network modules to evaluate the importance of bacteria between different states. By applying NetMoss to both simulated and real datasets, we demonstrate that it can efficiently reduce batch effects and identify more robust biomarkers that were neglected by traditional abundance-based methods. Furthermore, from a network perspective, we have found that, in pandisease microbiome studies, many gut bacteria are multidisease-related rather than disease-specific. The application of our network-based method greatly improves the efficiency of integrating multiple datasets and promotes the identification of microbial biomarkers for clinical diagnosis.

## Results

**Batch effect confounds integration of large-scale cohorts.** Multipopulation cohorts are currently widely used in the analysis of case–control studies; however, one of the most striking problems when integrating different datasets is the batch effect. On the one hand, different studies usually employ various experimental and computational methods during sample collection, processing and data generation, causing extensive biases in microbial profiles. On the other hand, taxon abundance varies substantially in different studies due to divergence in community composition and structure, which may lead to false interactions and network structures (Fig. 1a). For these reasons, direct integration of different datasets may cover the authentic characteristics of microbial communities and generate

[1]Beijing Institutes of Life Science, Chinese Academy of Sciences, Beijing, China. [2]University of Chinese Academy of Sciences, Beijing, China. [3]Key Laboratory of Systems Biology, Hangzhou Institute for Advanced Study, University of Chinese Academy of Sciences, Hangzhou, China. [4]State Key Laboratory of Integrated Management of Pest Insects and Rodents, Institute of Zoology, Chinese Academy of Sciences, Beijing, China. [5]These authors contributed equally: Liwen Xiao, Fengyi Zhang. ✉e-mail: zhfq@biols.ac.cn

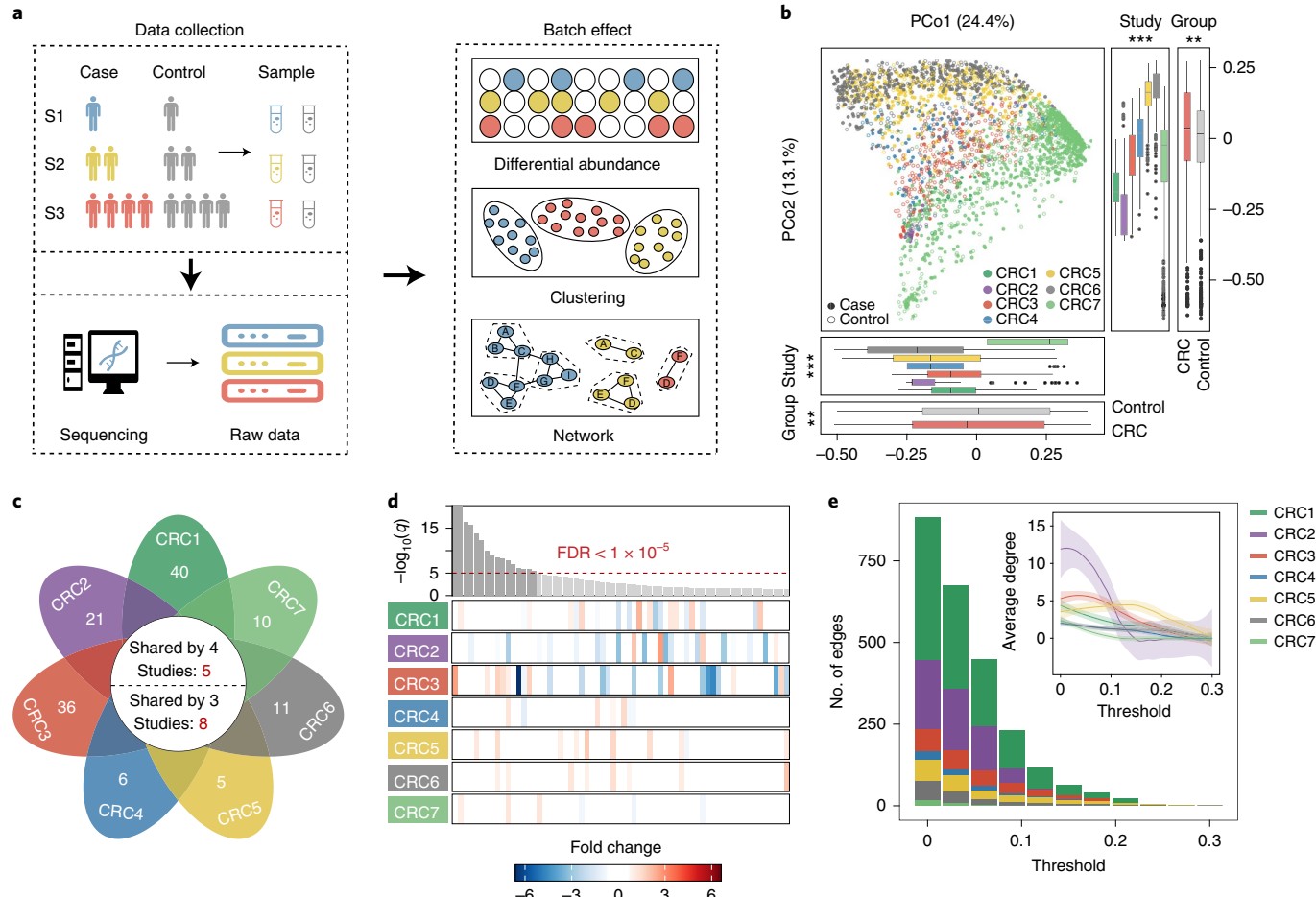

**Fig. 1 | Batch effects and challenges in meta-analysis of the gut microbiota. a**, Common challenges in the integration of multiple datasets. **b**, Principal coordinates analysis based on the relative abundance of control samples (open points) and CRC samples (filled points) from seven studies. Box plots represent differences among seven studies or between case and control groups. In the box plots: center line, median; box, interquartile range (IQR; the range between the 25th and 75th percentiles); whiskers, 1.5 × IQR; dots, outliers. Two-sided Wilcoxon test or Kruskal–Wallis rank sum test. ***$P < 0.001$; **$P < 0.01$; *$P < 0.05$. **c**, The number of differentially abundant bacteria with a two-sided Wilcoxon test in each study. The numbers on the leaves correspond to the unique differential bacteria of each study, and differential bacteria shared by multiple studies are shown in the central circle. **d**, Top: the bar height represents the meta-analysis significance of gut microbial genera derived from blocked Wilcoxon tests (top). Bottom: heatmap representing the fold change within individual studies. Bacteria are ordered by meta-analysis significance. **e**, The distribution of edges under different thresholds of microbial networks constructed from seven CRC studies. Inset: average degree under different thresholds; different colors of lines represent seven CRC studies, respectively. The gray regions indicate the 95% confidence intervals.

strong bias. Overall, to solve the problem of the integration of large microbiota datasets, more reliable approaches are urgently needed.

To evaluate potential bias in different microbiota datasets, we collected 2,742 gut microbiota datasets from seven independent colorectal cancer (CRC) studies representing three different countries (China, Germany and the United States; Supplementary Table 1). First, we explored the heterogeneity among different batches. Principal coordinates analysis (PCoA) indicated that the difference among studies was much greater than that between case–control groups (Kruskal–Wallis rank sum test, $P < 0.001$; Fig. 1b). Similar patterns were also observed in the results of the independent Wilcoxon rank sum test (Fig. 1c) and the blocked Wilcoxon test (Fig. 1d). Among all 665 genera from the seven CRC studies, although 142 were significantly different (false discovery rate (FDR) < 0.01), very few were shared by multiple studies. Specifically, only eight differential bacteria were shared by at least three studies, five by four studies, and none by more than five studies (Fig. 1c). Even for those shared bacteria, the alteration in abundance varied greatly in different studies (Fig. 1d). For example, the genus *Fusobacterium* was

significantly enriched in diseased individuals in most CRC studies but exhibited higher abundance in the healthy group in CRC2 (Fig. 1d; FDR < 0.01). In contrast, the genus *Lachnospira* was more abundant in the disease group in CRC2, but had higher abundance in the healthy group in other studies. Such discrepancies indicate that the conclusion is less convincing when ignoring batch effects during the integration of different cohorts.

We then attempted to explore the difference of network structure among different batches. Network topology in each CRC study was examined with different tools and different thresholds (Supplementary Fig. 1). Although the topological parameters of all seven studies tended to decrease as the threshold of correlation coefficients increased, the situation varied among studies (Fig. 1e). For example, when the threshold increased from 0 to 0.1, the number of edges in the CRC1 and CRC2 network decreased sharply while those in the other networks dropped more smoothly. The same tendency was observed in the variation of average degree (Fig. 1e), suggesting that the study size affects the topological structure of networks. In addition, small networks appear to be more sensitive

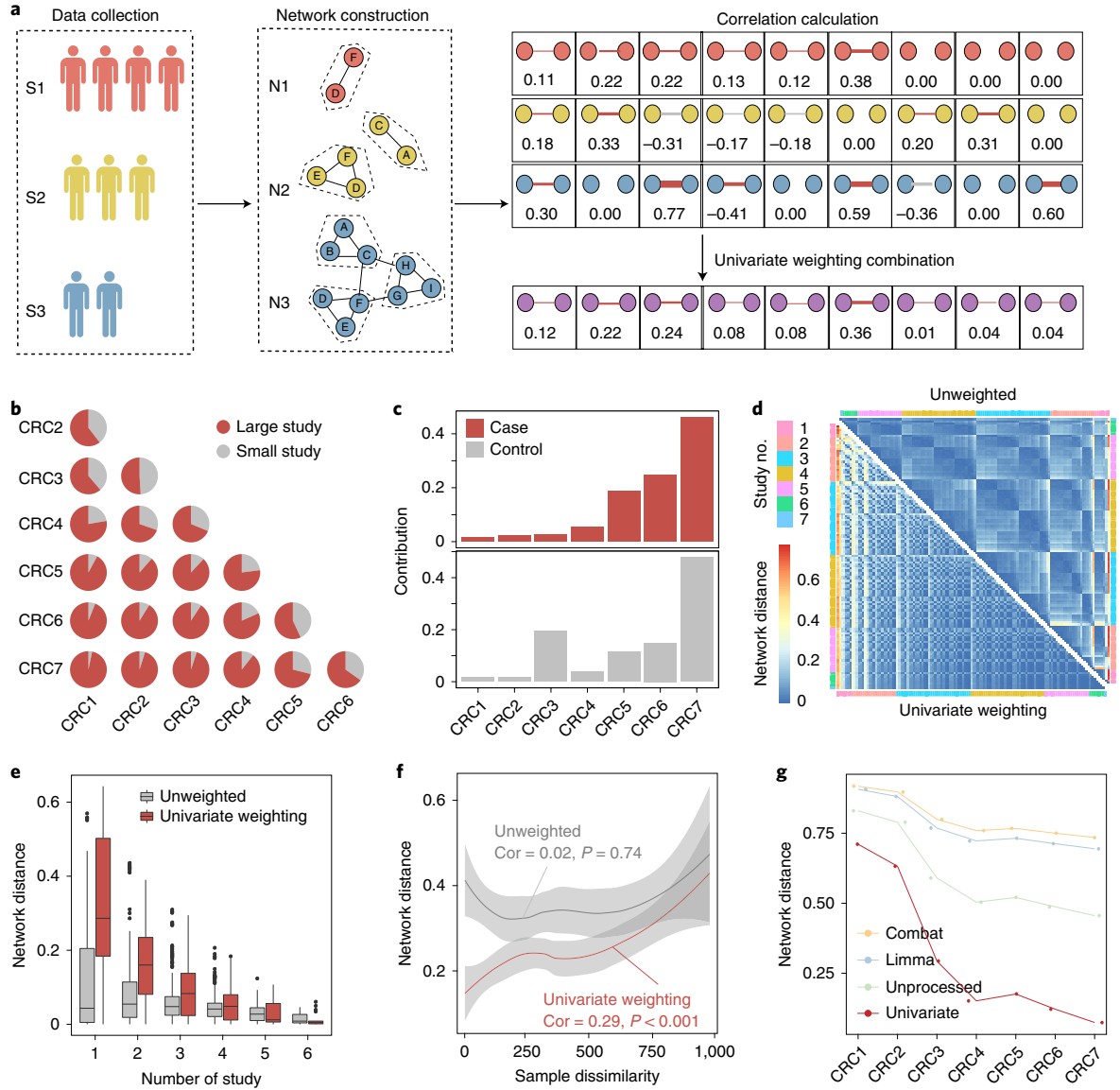

**Fig. 2 | Integrating multiple studies using a network-based method. a,** Workflow of network integration using the univariate weighting method. Different colors correspond to different studies. The number in the right panel corresponds to the correlation coefficients of each pair of bacteria in the networks. Zero corresponds to bacteria without interaction. **b,** Contribution rates of each CRC study in the combination of two study groups, in which red represents the larger study and gray the smaller study. **c,** Contribution of each study in the integrated network from the case (top) or control group (bottom). **d,** Pairwise network distance among 127 integrated networks using the univariate weighting method (lower left triangle) and unweighted method (upper right triangle). **e,** Network distance of integrated networks that include the same number of studies. In the box plots: center line, median; box, IQR (the range between the 25th and 75th percentiles); whiskers, 1.5×IQR; dots, outliers. **f,** Correlation between the distance of networks and difference of samples using the univariate weighting method (red line) and unweighted method (gray line), respectively. The shaded regions indicate the 95% confidence intervals. **g,** Network distance between seven CRC networks and the integrated network constructed using four different strategies.

to the choice of threshold. Consequently, the selection of different thresholds in network analysis may lead to different conclusions.

For clarity, we constructed co-occurrence networks of the seven studies. The results indicated that the microbial interactions were much weaker in large studies than in small studies (Supplementary Figs. 1 and 2). We speculate that microbial profiles in large cohorts are distributed more evenly; thus, the network structure is much looser than that in small communities. Accordingly, great differences were observed in the comparison of networks of different sizes when the threshold changed rapidly (Fig. 1e). Owing to the lack of appropriate normalization, neither the classic differential abundance method nor the previous integration network method

can achieve satisfactory performance for the integration of various microbiota datasets.

**Integration of networks using a univariate weighting method.** Considering that the networks constructed from large studies exhibit weak microbial interactions, directly integrating datasets with different sizes into one network might mask the real microbial features of large datasets. To address this, a univariate weighting method was introduced in our analysis, whereby a greater weight was assigned to the larger dataset to increase its contribution in the final integrated network (Fig. 2a). We first verified the method in a pairwise permutation test, in which any two networks were

integrated into one network. As shown in Fig. 2b, among all combinations of two study groups, the large study had a greater contribution to the integrated network, with the contribution increasing with the sample size of the included study. Similar results were observed in the integration of all seven networks: the larger the community size, the greater its contribution to the final integrated network (Fig. 2c), suggesting that this univariate weighting method can efficiently highlight the strength of large studies in the final network and reduce bias in the process of integration.

To further verify whether the univariate weighting method can remove batch effects in the process of integration, we permutated the seven studies to generate 127 different integrated networks (the number of studies included in the integrated networks ranged from one to seven). Compared to the situation with the traditional unweighted method, the distribution of network distance with the univariate weighting method not only exhibited a more even pattern (Fig. 2d), but also decreased more sharply with an increasing number of studies included in the integrated networks (Fig. 2e), demonstrating that integration of different datasets based on the univariate weighting method can reduce heterogeneity among studies. Notably, the univariate weighting method also showed a significantly higher correlation between network distance and sample dissimilarity (Fig. 2f; $P < 0.001$), suggesting its better performance in describing variation among studies. To explore the differences among different methods, we constructed networks using four different strategies: (1) integrating datasets simply based on abundance without removing batch effects (unprocessed); (2) integrating datasets based on the univariate weighting method; (3–4) integrating datasets based on abundance and removing batch effects using combat (3) or limma (4). Consistent with previous studies, the traditional methods showed inferior performance in batch effects removal on the microbial datasets (Supplementary Fig. 3). By contrast, the univariate weighting method showed a lower distance between the final integrated network and seven original networks, indicating its good performance in capturing original biological features (Fig. 2g).

**Prediction of transition using a network-based algorithm.** To delineate the transition from health to disease and to identify key bacteria during this process, we propose a NetMoss algorithm to perform network-based differential analysis (Fig. 3a; more details are provided in the NetMoss algorithm section in the Methods). We first generated two simulated networks to confirm that the NetMoss algorithm is able to measure variation in network structure between different states (Fig. 3b,c). After perturbation, 30 out of 40 submodules transitioned from module 1 to module 2, implying an alteration from health to disease (Fig. 3c). We then calculated the NetMoss scores of these 40 taxa in the integrated network to confirm whether our method can distinguish transited submodules from others. The results showed that a majority of transited submodules (86.7%) could be predicted by the NetMoss score (Fig. 3d), indicating its

great performance in identifying driver bacteria associated with state transition.

To further evaluate the performance of the NetMoss method, the Neighbor Shift (NESH) score[20] and the Jaccard Edge Index (JEI), which are both used to measure the variation of nodes in networks, were introduced to compare with our method. We re-perturbed the simulated network and added different noises to benchmark whether the three methods could identify transited submodules correctly. As shown in Fig. 3e, when random noises were added to taxon 81 to taxon 120, the NetMoss method outperformed the other two methods on distinguishing transited submodules from others. We then altered the noise level on the simulated networks and found that the area under the curve (AUC) of NetMoss remained high and stable (average AUC = 0.95; Fig. 3e, Supplementary Fig. 4 and Supplementary Table 2), further demonstrating its good performance and consistency on different community types. When perturbation occurs, the connection of bacteria changes as the structure of the network changes. Unlike NESH and JEI, the NetMoss algorithm not only takes node connection into consideration, but it also quantifies the node distance between different modules. Even a slight change in the network structure can be detected based on this module shift strategy, so the NetMoss method shows great advantages in the identification of biomarkers compared to other network-based methods.

**Identification of biomarkers in integrated CRC networks.** To identify disease-related bacteria, we integrated seven CRC studies into two integrated networks (case and control; Fig. 4a) and found that great differences existed between the case and control groups (Fig. 4b). For example, compared to the control group, Actinobacteria in the case group was greatly decreased, but Firmicutes was more abundant (Fig. 4b). In particular, in the small modules of the case group, the microbial composition was very simple, and among the four most common bacterial phyla, only Firmicutes was detected (Fig. 4b). Such distinctions indicated that the lack of certain bacteria in the microbial network may be associated with the transition from health to disease. We further retrieved 66 CRC-relevant bacteria from the gutMDisorder database[21] and found that the connection strength of marker bacteria was significantly higher than that of nonmarker bacteria in both the case and control network modules (Fig. 4c; $P < 0.001$, Wilcoxon test), suggesting the crucial role of these marker bacteria in integrated networks. Consequently, it would be an efficient strategy to determine disease-related bacteria from case–control network comparisons using the NetMoss method.

We then evaluated the accuracy of the NetMoss method using 66 known CRC-relevant bacteria, with 55 of them being present in the combined CRC datasets. A classic statistical test was used to identify differentially abundant bacteria between case and control groups, and the NetMoss score was used to assess the importance of bacteria in the integrated networks. Among the bacteria identified

**Fig. 3 | Identifying differential bacteria using the NetMoss algorithm. a**, Workflow of the NetMoss algorithm. **b**, Schematic of module transition between control and case networks. **c**, Module transition based on simulated networks. Random noise was generated and added to the control network (left), constituting the case network (right). The red bar corresponds to module 1 (m1), and the blue bar module 2 (m2). **d**, NetMoss score of each submodule during the transition of network states in **c**. The blue dotted line represents the threshold of the NetMoss score in the simulated networks. Submodules with NetMoss scores greater than the threshold correspond to differential network bacteria. Red dots represent submodules that experience transition, and gray dots represent submodules that do not transition. The shaded regions indicate the 95% confidence intervals. **e**, Comparison of submodule identification among three network-based methods. From top to bottom: NetMoss, NESH and JEI. The two columns on the left represent the density and distribution of submodule scores calculated using the three methods, respectively. In the leftmost column, positive nodes (red) correspond to transited submodules and negative nodes (gray) correspond to others. Gray areas in the taxon plots refer to the noise range added to the taxon. The third column shows the sensitivity (red) and specificity (blue) under different thresholds. The vertical blue dashed line corresponds to the best threshold to distinguish positive and negative nodes, which was determined based on the intersection point of sensitivity and specificity. The gray regions indicate the 95% confidence intervals. The rightmost column represents the prediction power of ten times of replication under different noise levels. Detailed AUC values are shown in Supplementary Table 2.

by the two methods, only 32% were marker bacteria using the statistical test method (FDR < 0.05); in contrast, 68% were successfully identified when NetMoss was implemented (NetMoss score > 0.12), suggesting that our network-based method substantially improves the efficiency of the identification of disease-related bacteria

(Fig. 4d,e). In particular, some genera showed a relatively high NetMoss score—for example, *Faecalibacterium*, *Actinetobacter* and *Parvionas* (NetMoss score > 0.99), which have been demonstrated to be associated with human health or disease (Fig. 4e). It should be noted that only 10 out of 43 taxa were identified by both

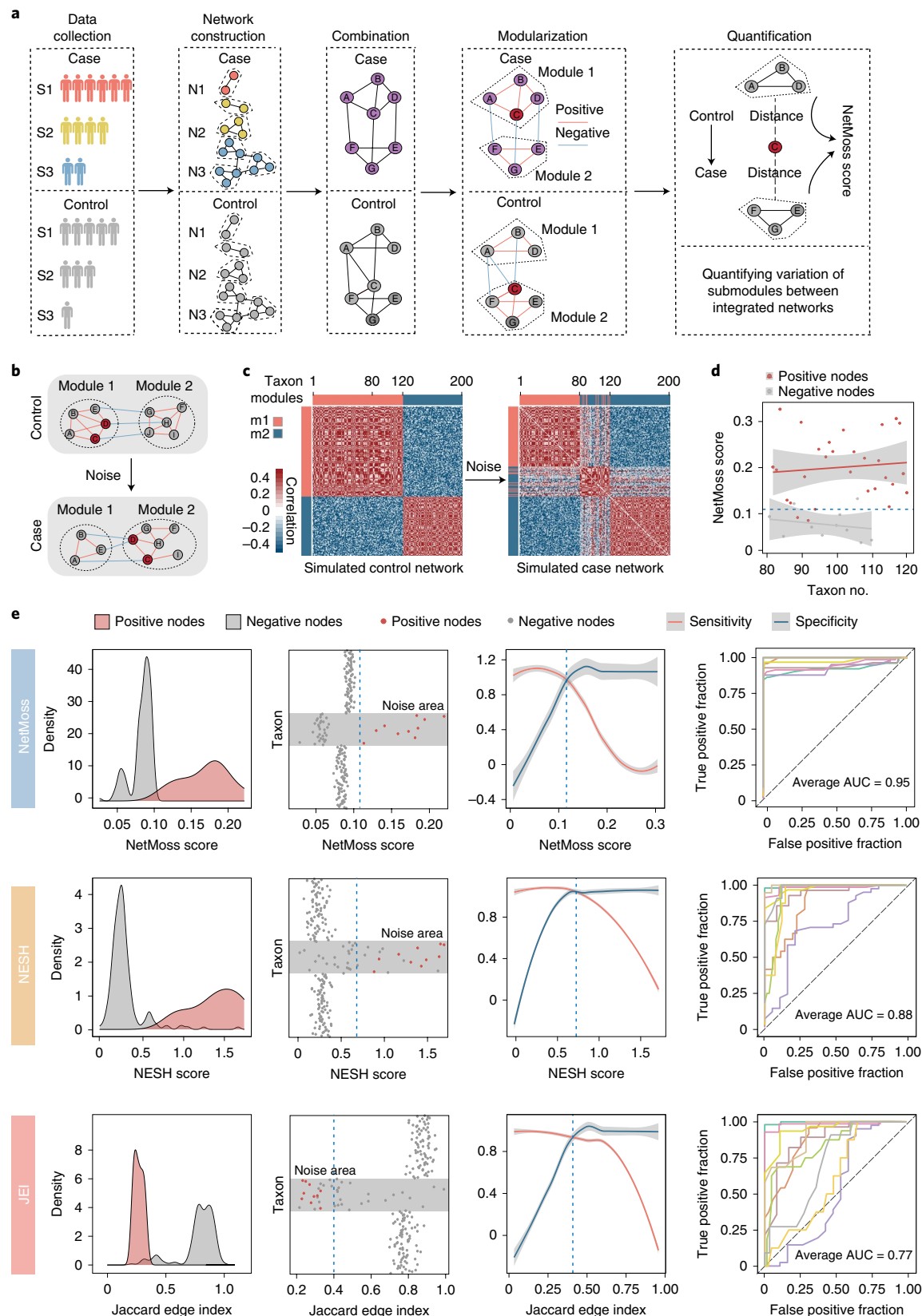

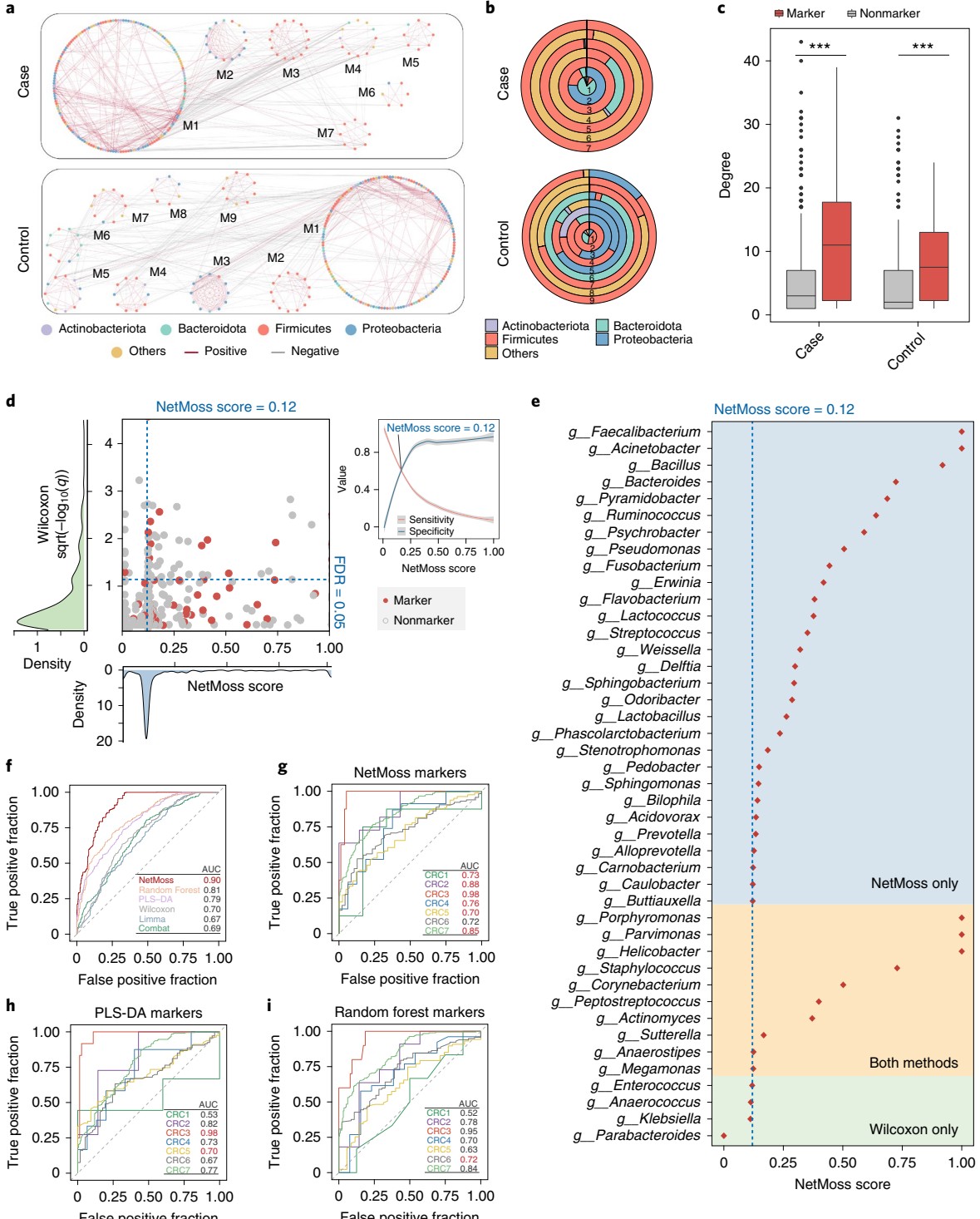

**Fig. 4 | Application of NetMoss for identifying CRC-associated bacteria. a**, Modules divided by the NetMoss algorithm in case (top) and control (bottom) groups. The color of dots represents bacteria at the phylum level, and the color of edges represents a positive correlation (red) or negative correlation (gray). **b**, Microbial composition in corresponding modules from the case (top) and control (bottom) groups. Different colors correspond to different phyla. **c**, Connection strength of each node in the integrated case and control network. Red represents microbial markers in the gutMDisorder database, and gray represents other bacteria. In the box plots: center line, median; box, IQR (the range between the 25th and 75th percentiles); whiskers, 1.5×IQR; dots, outliers. Two-sided Wilcoxon test. ***$P < 0.001$; **$P < 0.01$; *$P < 0.05$. **d**, Abundance-based significance or NetMoss score for each genus. Red dots represent microbial markers in the gutMDisorder database, and gray dots represent other bacteria. The two dashed blue lines indicate NetMoss score = 0.12 (vertical) and FDR = 0.05 (horizontal), respectively. The density plots represent the distribution of abundance-based significance (green) or NetMoss score (blue). The small panel on the right refers to sensitivity (red) and specificity (blue) under different thresholds, in which the point of intersection represents the best threshold to distinguish markers and nonmarkers. The gray regions indicate 95% confidence intervals. **e**, NetMoss score for markers identified by NetMoss only (blue area), Wilcoxon only (green area) or both methods (yellow area). **f**, Prediction power of six different methods on the combined CRC datasets. The red AUC value shows the best prediction across six methods. **g–i**, Prediction power of three methods (**g**, NetMoss; **h**, PLS-DA; **i**, Random Forest) on each of the seven separate studies. The red AUC value shows the best prediction of each study across three methods.

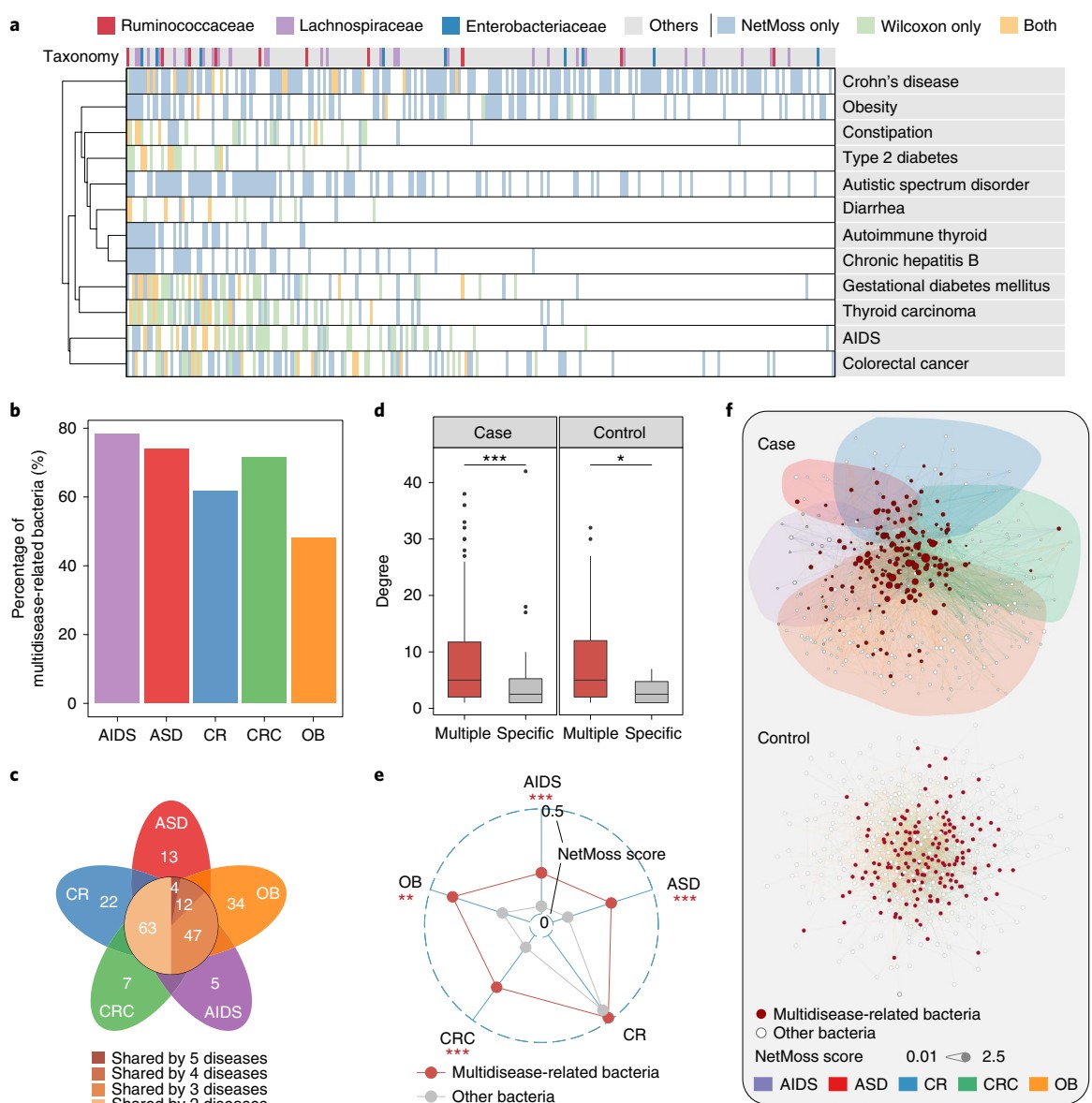

**Fig. 5 | NetMoss identifies bacteria associated with multiple diseases. a**, Differential bacteria identified by the NetMoss method only (blue), by the abundance-based method only (green) or by both methods (yellow). Only microbial families that contain over ten genera are shown in the color bar. **b**, The proportion of multidisease-related bacteria in the five most prevalent diseases. AIDS, acquired immunodeficiency syndrome; ASD, autistic spectrum disorder; CR, Crohn's disease; CRC, colorectal cancer; OB, obesity. **c**, The number of differential bacteria in each study. The number on the leaves corresponds to the unique differential bacteria of each disease, and differential bacteria shared by multiple diseases are shown in the central circle. **d**, The difference of network degree between multidisease-related bacteria (red) and disease-specific bacteria (gray) in case and control groups. In the box plots: center line, median; box, IQR (the range between the 25th and 75th percentiles); whiskers, 1.5 × IQR; dots, outliers. **e**, Average NetMoss score of multidisease-related bacteria (red) and disease-specific bacteria (gray) in five diseases. Two-sided Wilcoxon test. ***$P < 0.001$; **$P < 0.01$; *$P < 0.05$. **f**, The integrated network for healthy and diseased samples on combining five diseases. Each node represents one taxon and each edge represents the correlation between two taxa. The size of node corresponds to the NetMoss score, and the color of the node corresponds to multidisease-related bacteria (red) or other bacteria (open circles). The colors of edges represent different studies.

methods, indicating that the abundance-based method and the network-based method should complement each other in differential microbiota analysis.

To further explore the differences among the various methods, we examined the prediction power using six different strategies. The NetMoss group integrated datasets and identified markers using our network-based workflow (Fig. 3a), and the other five groups integrated datasets and identified markers based on abundance, two of which were further processed using combat or limma to remove batch effects. We observed that the efficiency of the traditional abundance-based method was very low, ranging from 16% to 25%, and most CRC markers could not be identified, no matter whether batch effects were removed or not (Supplementary Figs. 5 and 6a). By contrast, the NetMoss method exhibited a much higher AUC among these groups in both the combined and uncombined datasets (Fig. 4f–i and Supplementary Fig. 6b,c), demonstrating its robustness to different batches and its advantages in large-scale microbiome data integration. As well as at the genus level, the efficiency of NetMoss on both amplicon sequence variants (ASVs) and species levels was also robust (Supplementary Fig. 7).

In the CRC integrated networks, only 116 submodules (17.4%) changed between healthy and diseased groups in all modules, and such slight variations could not be recognized by other methods. The NetMoss algorithm, however, focuses on module shift and is more sensitive to perturbation between different networks.

**Application in pandisease microbiota studies.** Considering the complementary role of abundance-based and network-based methods in identifying disease-related bacteria, we further applied them to other diseases to determine the common characteristics of microbiota changes in these diseases. We analyzed 11,377 microbiota samples from public studies of different diseases (Supplementary Table 1) and found that, compared with the abundance-based method, the NetMoss method identified many more bacteria associated with disease (Fig. 5a). Intriguingly, these key bacteria exhibited two different patterns: some only correlated with a specific disease (disease-specific bacteria), whereas others exhibited wide associations with multiple diseases (multidisease-related bacteria; Fig. 5a). Unexpectedly, the latter accounted for the majority in all differential bacteria (Supplementary Fig. 8a). For example, many genera of Enterobacteriaceae and Lachnospiraceae, known as opportunistic pathogens, were found to be associated with infection and multiple diseases, such as CRC, diarrhea and type 2 diabetes[6,22,23]. We identified some of them as biomarkers in over five diseases (Fig. 5a and Supplementary Fig. 8b). In addition, several studies have reported strong associations between hepatitis B virus infection and *Streptococcus* or *Bacteroides*, which are also key bacteria in the occurrence of gestational diabetes mellitus[24–26]. Although a certain degree of abundance difference was observed between the case and control groups, most associations between diseases and these bacteria were only identified by using NetMoss (Fig. 5a).

We then focused on the five most prevalent diseases in the public datasets (Supplementary Table 1). We examined the prevalence of disease-specific bacteria and multidisease-related bacteria in each study and found that most biomarkers are multidisease-related bacteria (Fig. 5b). For example, four bacteria exhibited substantial differences between healthy and diseased groups in five diseases, but the number rose to 63 when only two diseases were included (Fig. 5c). Moreover, compared to disease-specific bacteria, multidisease-related bacteria were found to be much more abundant in both healthy and disease populations, confirming the importance of these biomarkers in the human gut.

To explore the role of multidisease-related bacteria in the development of disease, we compared the differences of network structure of five diseases. We found that multidisease-related bacteria showed a closer network connection and a higher NetMoss score compared to specific bacteria (Fig. 5d,e and Supplementary Fig. 8c; $P < 0.05$, Wilcoxon test). Such vital roles in the microbial networks suggested they may act as drivers in the development of multiple diseases. To further examine the association among multiple diseases, we integrated five disease networks into one combined network. Interestingly, unlike healthy controls, taxa from different diseases were largely separated from each other, with multidisease-related bacteria locating in the hub regions of the combined network (Fig. 5f and Supplementary Fig. 8e). Such opposite network structures between healthy and diseased groups further demonstrated the importance of microbial interaction networks in exploring the contribution of gut microbiota to various diseases.

## Discussion

Although some algorithms and tools have been developed to tackle the problem of batch effects in meta-analysis[27,28], most examine differential bacteria based on abundance; however, in human gut communities, microbes frequently interact with one another, forming a closely connected network[14]. Perturbation from outside may alter the structure of the network and change cooperative or competitive relationships among the bacteria. For this reason, the abundance of specific bacteria cannot describe the whole picture of the ecosystem, let alone the transition from health to disease. Network analysis has been widely used in various biological systems[16,29,30]. However, as the structure of the network is often associated with various factors, such as the size of datasets, selection of cutoffs and construction methods, it is difficult to compare networks from different studies directly. Consequently, integration of different networks is necessary to understand microbial interactions. Considering the robust characteristic of gut communities, a slight perturbation often imposes little effect on the whole structure of the microbial networks, and the distinction may only manifest as small variations in network submodules. Therefore, focusing on such variations of submodules may be a reasonable strategy to discriminate disease-associated bacteria from others.

CRC is one of the most common cancers across the world, and colorectal tumorigenesis is highly associated with gut microbial dysbiosis[31]. However, in contrast to genetic signatures, the gut microbiota associated with CRC is highly dependent on environmental factors such as diet and life style, which differ greatly in different countries, especially between western and non-western populations[32]. Such divergence poses a significant challenge to the early diagnosis of CRC based on microbial biomarkers and usually leads to contradictory results in different microbiome studies. By utilizing the NetMoss algorithm, we revealed the importance of *Lactobacillus* in the occurrence of CRC, which was demonstrated to have a protective effect for CRC[33,34], although it did not show a significant difference in abundance between case and control groups. This finding highlights the advantage of network-based methods for the identification of abundance-insensitive biomarkers. However, the NetMoss method still has some limitations. In practice, the clinical progress of CRC can be divided into several distinct stages characterized by different symptoms and divergent gut microbial compositions[35,36]. The state of patients, such as medicated or not, may also affect their gut microbial structures, resulting in inter-individual variation. However, such metadata are generally not available from public datasets, which makes it difficult to utilize such information in the NetMoss method. Taking detailed clinical factors into consideration will undoubtedly improve the accuracy of biomarker identification and may represent a new direction towards deep mining of clinical microbiome data.

The NetMoss method greatly facilitates the identification of significant biomarkers in the transition from health to disease and helps contribute to our understanding of the roles of the human microbiota in networks of ecosystems. With the integration of multiple cohorts based on this network-based algorithm, we believe that divergence among different studies can be largely reduced and that neglected details can be elucidated from a more comprehensive perspective.

## Methods

**Datasets.** We collected human gut microbiota datasets of different diseases from the National Center for Biotechnology Information (NCBI) to construct a multipopulation cohort. The keywords we searched in PubMed included 'gut microbiota', '16S', 'human', 'stool' and 'microbiome'. Only samples from adult stool were retained for downstream analyses. In total, 5,608 fecal samples of diseased individuals and 5,769 samples of healthy control individuals from 78 studies were collected in our research, covering 13 countries (Canada, China, Denmark, Finland, France, Germany, India, Italy, Japan, Mexico, Spain, Sweden and the United States). Among the 37 kinds of disease included in our research, CRC was the most prevalent, with 1,455 disease samples and 1,287 healthy samples. Accordingly, the processes of method development, validation and differential analysis were mainly based on CRC cohorts.

**Analysis of 16S rRNA sequences.** The raw data of 16S rRNA gene sequencing were analyzed using the QIIME2[37] platform (v2020.2). In brief, the DADA2 plugin was used to filter the sequencing reads and construct an ASVs feature table. The taxonomy information for the ASVs was assigned against the Silva Database (https://www.arb-silva.de) (v138.1) using the classify-sklearn algorithm in the feature classifier plugin. Low-abundance ASVs, whose relative abundance did not reach 0.1% in at least 10% of the samples, were excluded.

**Network analysis.** The co-occurrence network of microbes was constructed with SPIEC-EASI[38]. The topological coefficients of the network were calculated using the R package 'igraph'[39]. For each case or control group, different sizes of study were integrated into one network using the univariate weighting method to remove batch effects. Considering that the networks constructed from large studies exhibit weak co-abundance patterns, great weights were added to the large networks to increase their contribution to the final integrated network and thus reduce bias in the integration. The procedure was implemented as follows.

First, a co-occurrence network was constructed based on the abundance matrix of each study. Next, every pair of co-abundance patterns between any two taxa was aligned. The missing co-abundance patterns were filled with value 0. A univariate weighting method was implemented to add different weights to different pairs of co-abundance patterns based on the size of each study. During this process, the Hedges and Olkin method[40] was used to evaluate the conditional deviation of the correlation coefficient in each study. For a certain study $n_i$, the conditional deviation $v_i$ of correlation coefficient $r_i$ was calculated as

$$v_i = \frac{1 - r_i^2}{n_i - 1}$$

The weight of each pair of co-abundance patterns was defined as

$$\rho = \frac{\sum_{i=1}^{k} w_i r_i}{\sum_{i=1}^{k} w_i}$$

where $w_i$ is the reciprocal of $v_i$, and $k$ represents the number of studies.

To demonstrate real ecological processes, module division was conducted using WGCNA[41], with which microbes interacted cooperatively with one another in one single module while interacting competitively between any two modules. In the weighted networks, the connection strength of node $i$ was defined as the sum of the connections between this node and all other nodes in the network, as

$$k_i = \sum_{j=1}^{n} a_{ij}$$

where $a_{ij}$ represents the correlation coefficient between node $i$ and node $j$.

To highlight the importance of nodes in the network module structure, we redefined the connection strength of node $i$ as

$$k_i = \sum_{j=1}^{n_1} a_{ij} - \sum_{j=n_1+1}^{n} a_{ij}$$

where $n$ represents the number of all nodes in the network and $n_1$ represents the number of nodes inside a specific module.

**NetMoss algorithm.** The NetMoss score was used to measure the driving force of every node in the transition of the network structure. This was defined as follows. First, the correlation matrix of the health state was $A = [c_{ij}]$ and the correlation matrix of the disease state was $A' = \left[c'_{ij}\right]$, where $c_{ij}$ is the correlation coefficient of node $i$ and node $j$:

$$c_{ij} = \text{cor}(i, j)$$

To obtain the optimized module structure, linear transformation was implemented to convert $c_{ij}$ to $s_{ij}$

$$s_{ij} = \frac{1 + c_{ij}}{2}$$

Thus, the correlation matrices of the health and disease states after transformation were $B = [s_{ij}]$ and $B' = \left[s'_{ij}\right]$, respectively.

The soft threshold $\beta$ was calculated based on the WGCNA algorithm, and the weighted network $a_{ij}$ was

$$a_{ij} = \left|s_{ij}\right|^{\beta}$$

Thus, the weighted correlation matrices of health and disease states were $C = [a_{ij}]$ and $C' = \left[a'_{ij}\right]$, respectively.

The topological overlap matrix $\omega_{ij}$ of node $i$ and node $j$ was calculated as

$$\omega_{ij} = \frac{l_{ij} + a_{ij}}{\min\{k_i, k_j\} + 1 - a_{ij}}$$

where $l_{ij} = \Sigma_u a_{iu} a_{uj}$, $k_i = \Sigma a_{iu}$; $u$ represents other nodes besides node $i$ and node $j$ in the network.

The distance between node $i$ and node $j$ was defined as

$$d_{ij} = 1 - \omega_{ij}$$

Then, the distance matrices of the health and disease states were $D = [d_{ij}]$ and $D' = \left[d'_{ij}\right]$, respectively.

Module division was conducted in matrices $D$ and $D'$, with matrix $D$ containing $n$ modules and matrix $D'$ $m$ modules. The number of intersection modules of $D$ and $D'$ was $K$ ($K \leq mn$). The average distance of every node from intersection modules to health distance matrix $D$ and disease distance matrix $D'$ was calculated, obtaining $N$-by-$K$-order matrices $D_{\text{mod}}$ and $D'_{\text{mod}}$, respectively. The differential module distance matrix was defined as

$$\Delta D = D'_{\text{mod}} - D_{\text{mod}}$$

The NetMoss score of node $i$ in any intersection module is

$$\text{NMSS}(i)_{A \rightarrow B} = \sum_{j}^{\text{NeighborsA}} \Delta D_{ij} - \sum_{l}^{\text{NeighborsB}} \Delta D_{il}$$

where $A$ and $B$ represent the health and disease networks, respectively; NeighborsA represents all neighboring modules in the health network, and NeighborsB represents all neighboring modules in the disease network.

The intersection modules represent the stable elements during the transition from health to disease, where the transited modules resulted in alteration of the network structure. The NetMoss algorithm measures the driving force in the transition of the network structure to evaluate the importance of every node.

**Module shift stimulation and random noise production.** To verify the effect of the NetMoss algorithm in the identification of network structure, simulated networks were generated. Considering the sparsity of microbiota networks, we developed an algorithm to generate a simulated correlation matrix with a certain module structure and added random noise to the matrix to simulate natural disturbance.

The number of $g_k$ unit vectors was selected from vector space $R^{M_k}$ to construct an $M_k$-by-$g_k$ matrix $U_k$. The $k$th module was represented by a $g_k$-by-$g_k$ matrix $\Sigma_k$:

$$U_k = (u_1 | u_2 | \dots | u_k)$$

$$\sum_k = \rho_k \left( U_k^T U_k \right) + (1 - \rho_k) I$$

where $k$ represents the number of modules in the simulation matrix, $g_k$ represents the size of the $k$th module, $M_k$ represents the variation range of the correlation coefficient inside the $k$th module, $\rho_k$ represents the maximum correlation coefficient inside the $k$th module, and $I$ represents the unit vector.

Accordingly, an $N$-by-$N$ matrix was constructed, with modules $\Sigma_1, \Sigma_2, \dots, \Sigma_k$ arranged on the diagonal in order and the area outside the modules filled with 0.

To produce random noise, the number of $c$ unit vectors was selected from vector space $R^m$ to construct an $m$-by-$c$ matrix $U_k$. The random noise matrix $S$ was selected from a $c$-by-$c$ matrix $S_k$:

$$U_k = (u_1 | u_2 | \dots | u_c)$$

$$S_k = \varepsilon_k \left( U_k^T U_k \right)$$

$$\varepsilon_k = \varepsilon_1 + \frac{k - 1}{r - 1} (\varepsilon_r - \varepsilon_1)$$

where $r$ represents the row of the noise matrix, $c$ represents the column of the noise matrix, $m$ represents the variation range of noise, $\varepsilon_k$ represents $k$-order noise, in which $k = 1$ represents the minimum value of noise, and $k = r$ represents the maximum value of noise. Finally, random noise matrix $S$ was added to the corresponding module matrix to simulate natural disturbance.

**Statistical analysis.** All statistical analyses were conducted in R and visualized using the package 'ggplot2'. The blocked Wilcoxon test was applied using the R package 'coin'[42]. Principal coordinates analysis was implemented by the function 'pco' in the R package 'labdsv'. Co-occurrence networks were visualized using Cytoscape[43]. For all analyses regarding multiple comparisons, we used the FDR method to correct for multiple testing. Classification was applied using the combined markers, which were identified based on network (NetMoss) or abundance (the other five methods) in the combined CRC datasets. For the separate classification in each of the seven studies, the combined markers were screened using tenfold cross-validation before the prediction.

## Data availability
All sequence data supporting the findings of this study were obtained from Sequence Read Archive (SRA) of NCBI with the accession numbers listed in Supplementary Table 1. Source data are provided with this paper.

## Code availability

The source code for the NetMoss algorithm and data analysis scripts can be accessed at Zenodo[44]. NetMoss has been implemented as an R package, which can be accessed from GitHub (https://github.com/xiaolw95/NetMoss).

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

## Acknowledgements

This work was supported by grants from the National Natural Science Foundation of China (no. 32025009), the National Key R&D Program of China (nos. 2021YFA1301000 and 2021YFC2301300) and the Strategic Priority Research Program of the Chinese Academy of Sciences (no. XDB38020300).

## Author contributions

F. Zhao conceived the project. L.X. and F. Zhang designed the algorithm and performed the data analysis. L.X., F. Zhang and F. Zhao wrote the manuscript.

## Competing interests

The authors declare no competing interests.

## Additional information

**Correspondence and requests for materials** should be addressed to Fangqing Zhao.

