## [Peer Review File · Nature Computational Science]

Peer Review Information

Journal: Nature Computational Science

Manuscript Title: Large-scale microbiome data integration enables robust biomarker identification

Corresponding author name(s): Fangqing Zhao

Reviewer Comments & Decisions:

Decision Letter, initial version:

Date: 6th October 21 21:19:38

Last Sent: 6th October 21 21:19:38

From: jie.pan@us.nature.com

To: zhfq@biols.ac.cn

Subject: Decision on Nature Computational Science manuscript NATCOMPUTSCI-21-0795

Message: Dear Professor Zhao,

Thank you for submitting "Large-scale microbiome data integration enables robust biomarker identification" to Nature Computational Science, and we apologize for the delay in reaching a decision on your manuscript. Regretfully, we cannot offer to publish it in its current form.

Among the considerations that arise at this stage are the manuscript's likely interest to a broad range of researchers in computational science, the pressure on space for the various disciplines covered by Nature Computational Science, and the likelihood that a manuscript would seem of great topical interest to those working in the same or related areas of computational science. We do not doubt the technical quality of your work or that it will be of interest to others working in this area of research. However, I regret that we are unable to conclude that the paper provides the sort of substantial practical or conceptual advance that would be of immediate interest to a broad readership of researchers in computational science.

Should future experiments allow you to better demonstrate the advantage of your approach by providing detailed data comparisons against existing methods, we would be happy to look at a revised manuscript (unless, of course, something similar has by then been accepted at Nature Computational Science or appeared elsewhere). This includes submission or publication of a portion of this work somewhere else. In the case of eventual publication, the received date

would be that of the revised paper.

If you are interested in submitting a suitably revised manuscript in the future or if you have any questions, please contact me.

Thank you for your interest in Nature Computational Science. I am sorry that on this occasion we cannot be more positive.

Best regards,

Jie Pan, Ph.D.
Associate Editor
Nature Computational Science

** To transfer your manuscript to Communications Biology, or another Nature Portfolio journal, please use our <https://mts-natcomputsci.nature.com/cgi-bin/main.plex?el=A5DI1bT2A5FzZ2X6A9ftdeACvJZIMFFN2e1icH2yMbgZ> manuscript transfer portal. If you transfer to Nature journals or to the Communications journals, you will not have to re-supply manuscript metadata and files, unless you wish to make modifications. This link can only be used once and remains active until used.

All Nature Portfolio journals are editorially independent, and the decision on your manuscript will be taken by their editors. For more information, please see our http://www.nature.com/authors/author_resources/transfer_manuscripts.html?WT.mc_id=EMI_NPG_1511_AUTHORTRANSF&WT.ec_id=AUTHOR manuscript transfer FAQ page.

Note that any decision to opt in to In Review at the original journal is not sent to the receiving journal on transfer. You can opt in to *In Review* at receiving journals that support this service by choosing to modify your manuscript on transfer. In Review is available for primary research manuscript types only.

For Nature Research general information and news for authors, see <https://www.nature.com/nature-research/for-authors>.

Initial Decision After Appeal:

Date: 14th January 22 01:42:51
Last Sent: 14th January 22 01:42:51
Triggered By: Jie Pan
From: jie.pan@us.nature.com
To: zhfq@biols.ac.cn
BCC: jie.pan@us.nature.com
Subject: Decision on Nature Computational Science manuscript NATCOMPUTSCI-21-0795A-Z
Message: ** Please ensure you delete the link to your author homepage in this e-mail if you wish to forward it to your co-authors. **

Dear Professor Zhao,

Your manuscript "Large-scale microbiome data integration enables robust biomarker identification" has now been seen by 3 referees, whose comments are appended below. You will see that while they find your work of interest, they have raised points that need to be addressed before we can make a decision on publication.

The referees' reports seem to be quite clear. Naturally, we will need you to address all of the points raised.

While we ask you to address all of the points raised, the following points need to be substantially worked on:

- As suggested by referees, please use more recent pipeline and dataset for your study. All referees have concerns about the fact that the currently used pipeline or datasets are outdated.
- As suggested by referee #3, the benchmark and comparison with other methods need to be strengthened.
- Please improve your code files as suggested by referee #3.
- Please better discuss the limitations of NetMoss.
- Please improve the readability of your paper for more general audience by following the suggestions from referee #2

Please use the following link to submit your revised manuscript and a point-by-point response to the referees' comments (which should be in a separate document to any cover letter):

[REDACTED]

** This url links to your confidential homepage and associated information about manuscripts you may have submitted or be reviewing for us. If you wish to forward this e-mail to co-authors, please delete this link to your homepage first. **

To aid in the review process, we would appreciate it if you could also provide a copy of your manuscript files that indicates your revisions by making use of Track Changes or similar mark-up tools. Please also ensure that all correspondence is marked with

your Nature Computational Science reference number in the subject line.

In addition, please make sure to upload a Word Document or LaTeX version of your text, to assist us in the editorial stage.

To improve transparency in authorship, we request that all authors identified as 'corresponding author' on published papers create and link their Open Researcher and Contributor Identifier (ORCID) with their account on the Manuscript Tracking System (MTS), prior to acceptance. ORCID helps the scientific community achieve unambiguous attribution of all scholarly contributions. You can create and link your ORCID from the home page of the MTS by clicking on 'Modify my Springer Nature account'. For more information please visit www.springernature.com/orcid.

We hope to receive your revised paper within three weeks. If you cannot send it within this time, please let us know.

Best regards,

Jie Pan, Ph.D.
Associate Editor
Nature Computational Science

Reviewers comments:

Reviewer #1 (Remarks to the Author):

The authors propose a novel algorithm called NetMoss to identify robust biomarkers associated with diseases. Evaluated by the simulations and real datasets, they showed that NetMoss performed well in the identification of disease-associated biomarkers and could be a complement of the abundance-based differential analysis method.

The following are my comments and questions:

1. The authors generate 97% sequence identity OTUs using QIIME. QIIME 2 succeeded QIIME 1 as of 2018 and QIIME 1 is no longer supported. Besides, currently most researchers generate ASVs (Amplicon Sequence Variants) that take into account sequencing error profiles to improve specificity and sensitivity for taxonomic identification. I would suggest processing the data using DADA2 in QIIME 2.
2. The Greengenes database hasn't been updated since 2013. SILVA or other recently developed database is preferred.
3. What are the unprocessed data referred to? The raw counts? I don't think it's reasonable to directly use the raw counts for the downstream analysis. Some simple normalizations such as turning it to relative abundance would help. If the authors did some processes for the 'unprocessed data', please point them out.

4. The authors used SparCC to build the networks. In previous studies (<http://dx.doi.org/10.1038/ismej.2015.235>), SparCC was shown to have higher type I error than the specified p-value. Therefore, the network build from SparCC may have many errors. I would suggest to consider other network building tools such as LSA. It is not clear from the writing whether the value r_i was calculated based on SparCC or the correlation coefficient of the relative abundance. Please specify.
5. Page 15, line 1 above the equation for v_i , change "for a certain study n_i " to "for a certain study containing n_i subjects". For the equation of v_i , I think the power "2" for $(1-r_i^2)$ is incorrect and should be deleted. Please double check the reference and cite the reference if you are sure of the equation. In several resources, the standard error of r_i was given as $\sqrt{(1-r_i^2)/(n-2)}$. https://en.wikipedia.org/wiki/Pearson_product-moment_correlation_coefficient#Inference.
6. In the next equation for ρ , it is the weighted average of all the correlation coefficients across the studies. However, when we take the weighted average, the weights should usually be the inverse of the standard deviation, instead of the variance. In this manuscript, the weights were the inverse of the variance. Please give reasons why such weights were chosen.
7. Figure 2g, what's the reason for significantly lower network distance after implementing the univariate weighting method of CRC2 data? Other studies had similar distances using the 4 methods you compared.
8. Please add more details about the value of simulation parameters. The variation range M_k and the maximum correlation coefficient ρ_k are needed.
9. No repetitions for the simulations? It didn't convince me if you only did it one time.
10. How did you choose the threshold for NetMoss score?
11. Some small slips: p17, Σk should be Σ_k ; p23, legend of figure 3d, red dotted line should be blue dotted line.

Reviewer #2 (Remarks to the Author):

In this manuscript, Xiao et al described a large co-abundance network-based microbiota analysis by applying a newly proposed algorithm NetMoss to colorectal cancer (CRC) from 7 cohorts with varied sizes as a case study and expanding to 5 diseases. The co-abundance networks were constructed from genus-level information. The authors pointed out batch effect among studies contribute highly to the contradictory outcomes in microbiome analysis. The NetMoss, as a univariate weighting method, where the authors assigned more weight to larger datasets, demonstrated better performance than previous reported tools, such as combat and limma, in capturing original biological relationships in both simulated and collected datasets from public resources. By utilizing the NetMoss, the authors identified highly prevalent microbiome features that are widely associated to multiple diseases in

populations as well as disease-specific taxa, which in general accounted for less in the differential microbiome features. The authors finally strengthened the importance of microbial co-abundance networking shifts in dissecting the contribution of gut microbiota to various diseases.

The study is a large descriptive undertaking and makes use of existing 16S rRNA sequencing datasets from freely available cohorts. The employed algorithms and statistical approaches seem appropriate; however the manuscript lacks in my view the necessary clarity and scrutiny on physiological relevance of the findings. The reviewer understood the manuscript at this field should be written in a very technical way, but the rare interpretation in outcomes from the analysis will limit the readership of this manuscript. The authors are strongly recommended to re-write the discussion section, in particular the paragraph discussing the CRC-relevant biomarkers. The authors may consider composing this discussion in a bio-medically translated manner rather than citing the results from relevant descriptive studies.

The entire study is based on the microbiome features at genus level using 16S rRNA sequencing as key words for cohort searching. However, the gut bacterial genus is a broad concept and lacks specificity for the interpretation of analysis outcomes. In this study, the reviewer thinks the low resolutions in taxonomic annotations may lead to the limited identification of disease-specific features, as most of the pandemic-associated bacterial genus in this study are high abundant and prevalent in human gut microbiome. How did the authors decide to use genus datasets for co-abundance network construction? What would happen if the resolution reached bacterial species? There are many metagenomics datasets that are available and valuable to be explored by this algorithm. Moreover, the authors may also consider testing the NetMoss algorithm on microbial functional (pathway) modules from gut metagenomic sequencing datasets.

The clinical attribute CRC is inappropriate if only used alone. The authors clearly must try to discriminate between benign (stage 0) and malignant (III or IV) stages if metadata are available for the included datasets.

Also, in the cross-sectional cohorts, patients were usually medicated, which has been reported impacting gut microbial structures. Therefore, the medication may remodel or regulate the co-abundance networks and mask the module shifts induced by disease. The reviewer understood the difficulties in capturing the network alterations from cross-sectional studies. But it will be valuable if the authors include this point to the limitations of this study.

Reviewer #3 (Remarks to the Author):

This work proposes a new method for biomarker detection in microbiota profiling studies. The NetMoss method is based on the analysis of taxonomic co-abundance networks, also called network modules, and their association with the endpoint variable. The method detects network topology to reduce batch effects for added robustness in biomarker identification. A unique feature is that NetMoss focuses on shifts in network modules rather than on variation in bacterial abundance.

Performance gains are shown in comparison with standard alternatives. In addition, the authors analyse multiple public data sets and report previously overlooked CSC biomarkers and biomarkers that seem to be shared across many common diseases.

Overall, the reporting is fluent and easy to follow, the language is good quality. The methodology is based on widely used approaches, takes into account the batch effects

and compositionality biases, and appropriate references have been provided. There are some shortcomings that could be improved in particular in terms of benchmarking and code availability.

Major

1. Benchmarking tests with real data include limma, combat, and "unprocessed" methods. However, widely used methods for such classification tasks in the contemporary taxonomic profiling studies include random forest, PLS-DA, and xgboost. It would be important to include comparisons with some, or preferably all of these methods. They are all available as R packages, and their use in the similar classification context is straightforward.
2. The 16S rRNA analysis pipeline seems to be based on relatively old methodology; the QIIME reference (44) is to the 2010 paper, whereas there is a newer version QIIME2 from 2019 (DOI: 10.1038/s41587-019-0209-9). Also the GreeGenes database has not been updated after 2013 to my knowledge, and is currently severely outdated. The OTUs are nowadays often replaced with ASVs from DADA2 pipeline (included in QIIME2) since this offers a better resolution. It would be important to evaluate how much this influences the current results, and preferably the work should be updated to use more up-to-date software and databases.
3. It would be useful to see examples and/or discussion on cases where the NetMoss method has shortcomings. This will help to understand the limitations of the method more deeply.
4. Data and code availability. The analyses are based on simulations and publicly available data sets, all data is available. The source code is available through github but could be improved as follows:
 - a) Add permanent DOI through e.g. Zenodo. This guarantees that the exact code version used in this manuscript will be preserved permanently. See <https://guides.github.com/activities/citable-code/>
 - b) In addition to the source code of the method, the repository should include code that was used to create the figures included in the manuscript. I did not find this information from the README or browsing the files quickly.
 - c) The source code is missing license, hence it is not clear if the code is openly available (i.e. with an open license). Consider adding open source license on the code as is often recommended (see e.g. 10.1371/journal.pcbi.1002598)
 - d) If the R scripts could be implemented as an R package, the method would be easier to use. The lack of this will remarkably limit the potential user base, and makes benchmarking with alternatives more difficult.

e) Consider including Rmarkdown vignette in the code that shows how to use the tools.

Minor

The authors refer to "microbial interactions" but the taxonomic co-abundance networks are statistical associations rather than biological interactions, although in some cases these overlap. The abundance data itself does not, however, differentiate between statistical and biological interactions. I suggest to remove references to microbial interactions when discussing the methodology, and instead systematically use the term network module, co-occurrence, co-abundance patterns, or a similar term.

p5: tropological -> topological

SparCC has been used for taxonomic network detection. The SPIEC-EASI is a bit newer one and has shown considerable speed improvements with the same overall performance. It would be good to cite SPIEC-EASI in Discussion as an alternative method, and consider implementing that into the workflow.

Author Rebuttal to Initial comments**Editorial comments:**

While we ask you to address all of the points raised, the following points need to be substantially worked on:

- As suggested by referees, please use more recent pipeline and dataset for your study. All referees have concerns about the fact that the currently used pipeline or datasets are outdated.

Response: As suggested by the reviewers, DADA2 in QIIME2 and a much newer database SILVA was used to reanalyze our datasets. All the figures and the results have been updated based on the new abundance tables accordingly. Please refer to our responses to the comments 1-2 by the 1st reviewer for details.

- As suggested by referee #3, the benchmark and comparison with other methods need to be strengthened.

Response: As suggested, random forest and PLS-DA methods have been used to compare with our network-based method. Both on the combined datasets and each separated dataset, our NetMoss method outperformed previous methods. Please refer to our responses to the comment 1 by the 3rd reviewer for details.

- Please improve your code files as suggested by referee #3.

Response: A new version of source code has been deposited in ZENODO with doi 10.5281/zenodo.5913042. The R package implemented in our method can be accessed at <https://github.com/xiaolw95/NetMoss>. The source code has included a license and is open to the public. Please refer to our responses to the comment 4 by the 3rd reviewer for details.

- Please better discuss the limitations of NetMoss.

Response: The limitations of NetMoss have been discussed in our revised manuscript. (Page 14, Line 22-31)

- Please improve the readability of your paper for more general audience by following the suggestions from referee #2

Response: We have revised the Discussion to make it more favorable for the readers (Page 13-14).

Review comments:

Reviewer #1 (Remarks to the Author):

The authors propose a novel algorithm called NetMoss to identify robust biomarkers associated with diseases. Evaluated by the simulations and real datasets, they showed that NetMoss performed well in the identification of disease-associated biomarkers and could be a complement of the abundance-based differential analysis method.

Response: We greatly appreciate the reviewer's comments on the novelty and significance of our study. All the comments raised by the reviewer have been seriously considered, and the manuscript has been extensively revised based on these comments.

The following are my comments and questions:

1. The authors generate 97% sequence identity OTUs using QIIME. QIIME 2 succeeded QIIME 1 as of 2018 and QIIME 1 is no longer supported. Besides, currently most researchers generate ASVs (Amplicon Sequence Variants) that take into account sequencing error profiles to improve specificity and sensitivity for taxonomic identification. I would suggest processing the data using DADA2 in QIIME 2.

Response: Thank you for pointing this out. As suggested, QIIME2 was used to generate ASVs for the downstream analysis and all the results have been updated in this revised manuscript.

2. The Greengenes database hasn't been updated since 2013. SILVA or other recently developed database is preferred.

Response: Thank you for this suggestion. In this revised manuscript, we used SILVA to replace Greengenes in the taxonomic identification. The results have been updated based on the new abundance tables accordingly.

3. What are the unprocessed data referred to? The raw counts? I don't think it's reasonable to directly use the raw counts for the downstream analysis. Some simple normalizations such as turning it to relative abundance would help. If the authors did some processes for the 'unprocessed data', please point them out.

Response: We are sorry for the confusion of our statement. The “unprocessed data” refers to the dataset directly integrated from different studies based on abundance without the removal of batch effects. The “unprocessed” is an opposite of “limma processed” or “combat processed”, which have removed batch effects using either limma or combat. In fact, although batch effects are not removed in the “unprocessed dataset”, it still has been normalized to relative abundance. We have clarified this point in the manuscript (Page 6 Line 33-35). To avoid confusion, we have changed it to “Wilcoxon” (**Figure 4g**).

4. The authors used SparCC to build the networks. In previous studies (<http://dx.doi.org/10.1038/ismej.2015.235>), SparCC was shown to have higher type I error than the specified p-value. Therefore, the network build from SparCC may have many errors. I would suggest to consider other network building tools such as LSA. It is not clear from the writing whether the value r_i was calculated based on SparCC or the correlation coefficient of the relative abundance. Please specify.

Response: We greatly appreciate the reviewer for this kind suggestion. As mentioned by the reviewer, SparCC has higher type I errors. Since LSA is more suitable for time series data, another tool called SPIEC-EASI, which has shown considerable improvement on the network construction¹, was used to build networks in this revised manuscript. Based on the new network, some results of our study indeed have improved. For example, the contribution of each network in the integrating process was more relevant to the sample size (**Figure 2b-2c**) and the variation of network distance among different studies was closer to the real situations (**Figure 2e-2g**).

Besides the SPIEC-EASI method, we further compared the influence of different network construction methods on marker identification. As shown in **Re. Fig. 1**, networks constructed by SPIEC-EASI were much sparser than those by the other two methods. In contrast, Pearson-based method tended to build networks with more positive correlation edges (**Re. Fig. 1a-1b**). Although different methods lead to different network topologies, the markers identified by different networks are largely overlapped (**Re. Fig. 1c**). Especially when using SPIEC-EASI, 84% of markers were shared by other tools (**Re. Fig. 1c**). The high overlap of SPIEC-EASI with other methods and great improvement of its application in network integration indicate the accordance with the real biological features, and thus this strategy was

chosen to build network in our study.

The value r_{r} was calculated based on the network constructed by SPIEC-EASI.

Re. Fig. 1 Comparison of different network construction methods. a, Topological parameters of seven CRC microbial networks in the case group ($P < 0.01$). b, Topological parameters of seven CRC microbial networks in the control group ($P < 0.01$). c, The overlap of markers identified by NetMoss based on different network construction methods.

Reference

1. Kurtz ZD, Müller CL, Miraldi ER, Littman DR, Blaser MJ, Bonneau RA (2015) Sparse and Compositionally Robust Inference of Microbial Ecological Networks. *PLoS Comput Biol* 11(5): e1004226. doi:10.1371/journal.pcbi.1004226

5. Page 15, line 1 above the equation for s_{v_i} , change “for a certain study n_i ” to “for a certain study containing n_i subjects”. For the equation of s_{v_i} , I think the power “2” for $(1-r_i^2)$ is incorrect and should be deleted. Please double check the reference and cite the reference if you are sure of the equation. In several resources, the standard error of r_i was given as $\sqrt{(1-r_i^2)/(n-2)}$.

https://en.wikipedia.org/wiki/Pearson_productmoment_correlation_coefficient#Inference.

Response: As suggested, we have revised the equation and corresponding description. The power “2” for $(1-r_i^2)$ has been deleted (Page 16 Line 5).

6. In the next equation for ρ , it is the weighted average of all the correlation coefficients across the studies. However, when we take the weighted average, the weights should usually be the inverse of the standard deviation, instead of the variance. In this manuscript, the weights were the inverse of the variance. Please give reasons why such weights were chosen.

Response: Thank you for this suggestion. As suggested, we have revised the weight for the inverse of the standard deviation (Page 16 Line 5).

7. Figure 2g, what’s the reason for significantly lower network distance after implementing the univariate weighting method of CRC2 data? Other studies had similar distances using the 4 methods you compared.

Response: The significantly lower network distance for CRC2 data may be derived from spurious noises during the network construction. It has been corrected by using a new network construction method (SPIEC-EASI) (Figure 2g).

8. Please add more details about the value of simulation parameters. The variation range M_k and the maximum correlation coefficient ρ_k are needed.

Response: All simulation parameters have been provided in the script of Figure 3, which can be accessed at <https://doi.org/10.5281/zenodo.5913042>. For different repetitions of the simulation, the parameter M_k and the ρ_k was different. For Figure 3b and 3c, the variation range of modules M_k was from 81 to 120, and the maximum correlation coefficient ρ_k was set to 0.5.

9. No repetitions for the simulations? It didn't convince me if you only did it one time.

Response: We are sorry for the unclear statement in our manuscript. In fact, ten times of repetition have been performed during the simulation step, the results were shown in **Figure 3e** (the rightmost roc) and **Supplementary Figure 5**, which indicate that NetMoss shows a stable and superior performance over previous methods.

10. How did you choose the threshold for NetMoss score?

Response: The threshold can be manually set by the users. Basically, the greater the NetMoss score is, the more significantly the taxon differs between case and control groups. By default, the threshold of NetMoss score was determined based on the intersection point of sensitivity and specificity.

11. Some small slips: p17, Σk should be Σ_k ; p23, legend of figure 3d, red dotted line should be blue dotted line.

Response: We greatly appreciate the reviewer for pointing this out. They have been corrected accordingly (Page 18 Line 9; Page 23 Line 11).

Reviewer #2 (Remarks to the Author):

In this manuscript, Xiao et al described a large co-abundance network-based microbiota analysis by applying a newly proposed algorithm NetMoss to colorectal cancer (CRC) from 7 cohorts with varied sizes as a case study and expanding to 5 diseases. The co-abundance networks were constructed from genus-level information. The authors pointed out batch effect among studies contribute highly to the contradictory outcomes in microbiome analysis. The NetMoss, as a univariate weighting method, where the authors assigned more weight to larger datasets, demonstrated better performance than previous reported tools, such as combat and limma, in capturing original biological relationships in both simulated and collected datasets from public resources. By utilizing the NetMoss, the authors identified highly prevalent microbiome features that are widely associated to multiple diseases in populations as well as disease-specific taxa, which in general accounted for less in the differential microbiome features. The authors finally strengthened the importance of microbial co-abundance networking shifts in dissecting the contribution of gut microbiota to various diseases.

Response: We greatly appreciate the reviewer's comments on the novelty and significance of our study. All the comments raised by the reviewer have been seriously considered, and the manuscript has been extensively revised based on

these comments.

The study is a large descriptive undertaking and makes use of existing 16S rRNA sequencing datasets from freely available cohorts. The employed algorithms and statistical approaches seem appropriate; however the manuscript lacks in my view the necessary clarity and scrutiny on physiological relevance of the findings. The reviewer understood the manuscript at this field should be written in a very technical way, but the rare interpretation in outcomes from the analysis will limit the readership of this manuscript. The authors are strongly recommended to re-write the discussion section, in particular the paragraph discussing the CRC-relevant biomarkers. The authors may consider composing this discussion in a bio-medically translated manner rather than citing the results from relevant descriptive studies.

Response: Thank you for these valuable and constructive comments. In this revised manuscript, we have recomposed our manuscript to improve the readability for more general audience and have added more biomedically translated discussion (Page 13 Line 24-35; Page 14 Line 1-3). As largely reserved biological features, the performance of NetMoss is better than previous tools.

The entire study is based on the microbiome features at genus level using 16S rRNA sequencing as key words for cohort searching. However, the gut bacterial genus is a broad concept and lacks specificity for the interpretation of analysis outcomes. In this study, the reviewer thinks the low resolutions in taxonomic annotations may lead to the limited identification of disease-specific features, as most of the pandemic-associated bacterial genus in this study are high abundant and prevalent in human gut microbiome. How did the authors decide to use genus datasets for co-abundance network construction? What would happen if the resolution reached bacterial species? There are many metagenomics datasets that are available and valuable to be explored by this algorithm. Moreover, the authors may also consider testing the NetMoss algorithm on microbial functional (pathway) modules from gut metagenomic sequencing datasets.

Response: We agree with the reviewer on this comment. Considering that most previous studies on gut microbiota were based on 16S rRNA sequencing, we mainly focused on the genus-level marker identification in our study, which may have limitations on identifying disease-specific features. In this revised manuscript, we added more shotgun-based metagenomic sequencing datasets and also

reanalyzed the 16S rRNA datasets at the OTU level. As shown in **Re. Fig. 2**, the performance of NetMoss significantly improved on the higher resolution metagenomic datasets, indicating the robustness of our network-based method and its consistency on different taxonomic levels.

Re. Fig. 2 Marker identification from different taxonomic levels. a, Markers identified by NetMoss at the OTU level. b, markers identified by NetMoss at the species level. c, Prediction power at three different taxonomic levels (genus level and OTU level from 16S dataset, and species level from metagenome dataset).

The clinical attribute CRC is inappropriate if only used alone. The authors clearly must try to discriminate between benign (stage 0) and malignant (III or IV) stages if metadata are available for the included datasets. Also, in the cross-sectional cohorts, patients were usually medicated, which has been reported impacting gut microbial structures. Therefore, the medication may remodel or regulate the co-abundance networks and mask the module shifts induced by disease. The reviewer understood the difficulties in capturing the network alterations from cross-sectional studies. But it will be valuable if the authors include this point to the limitations of this study.

Response: Thank you for these comments. We understand the reviewer's concerns. As most public datasets did not provide metadata information related to the stage of CRC or the medication records of patients, it is very hard for us to consider such clinical information in our method. As a matter of fact, the integration with biological features is a challenging task for most existing tools. We have added this point to the limitations of this study in our revised manuscript (Page 14 Line 22-31).

Reviewer #3 (Remarks to the Author):

This work proposes a new method for biomarker detection in microbiota profiling studies. The NetMoss method is based on the analysis of taxonomic co-abundance networks, also called network modules, and their association with the endpoint variable. The method detects network topology to reduce batch effects for added robustness in biomarker identification. A unique feature is that NetMoss focuses on shifts in network modules rather than on variation in bacterial abundance. Performance gains are shown in comparison with standard alternatives. In addition, the authors analyse multiple public data sets and report previously overlooked CSC biomarkers and biomarkers that seem to be shared across many common diseases.

Overall, the reporting is fluent and easy to follow, the language is good quality. The methodology is based on widely used approaches, takes into account the batch effects and compositionality biases, and appropriate references have been provided. There are some shortcomings that could be improved in particular in terms of benchmarking and code availability.

Response: We greatly appreciate the reviewer's comments on the novelty and significance of our study. All the comments raised by the reviewer have been seriously considered, and the manuscript has been extensively revised based on these comments.

Major

1. Benchmarking tests with real data include limma, combat, and "unprocessed" methods. However, widely used methods for such classification tasks in the contemporary taxonomic profiling studies include random forest, PLS-DA, and xgboost. It would be important to include comparisons with some, or preferably all of these methods. They are all available as R packages, and their use in the similar classification context is straightforward.

Response: We thank the reviewer for this suggestion. As suggested, random forest and PLS-DA methods have been compared with our network-based method (**Figure 4g** and **Re. Fig. 3**). Among all these six different methods, NetMoss showed the highest AUC on both combined datasets (AUC = 0.84, **Figure 4g**) and each separated dataset (**Re. Fig. 3a**). The classic tools, such as random forest and PLS-DA, performed well in the overall classification between healthy and diseased samples (AUC > 0.7, **Figure 4g**). However, the efficiency of both methods was not consistent in different studies (**Re. Fig. 3b-3c**). Especially for dataset with small sample size (e.g., CRC1, n = 46) which lacks enough training features, random forest and PLS-DA showed much poorer performance. For example, 41 unique markers were only found in CRC1, and among 159 CRC markers identified by random forest, only 22 (13.8%) was shared by all seven studies (**Re. Fig. 3d- 3e**). Compared with them, NetMoss showed a more stable and better performance in different studies (**Re. Fig. 3a**), demonstrating its robustness to different batches and its advantages on the large-scale microbiome data integration.

Re. Fig. 3 Classification comparison of different methods in seven CRC studies. a, Prediction power of NetMoss in seven CRC studies. b, Prediction power of PLS-DA in seven CRC studies. c, Prediction power of random forest in seven CRC studies. d, Unique markers in seven CRC studies identified by random forest. e, The overlap of markers identified by random forest.

2. The 16S rRNA analysis pipeline seems to be based on relatively old methodology; the QIIME reference (44) is to the 2010 paper, whereas there is a newer version QIIME2 from 2019 (DOI:10.1038/s41587-019-0209-9). Also the GreeGenes database has not been updated after 2013 to my knowledge, and is currently severely outdated. The OTUs are nowadays often replaced with ASVs from DADA2 pipeline (included in QIIME2) since this offers a better resolution. It would important to evaluate how much this influences the current results, and preferably the work should be updated to use more up-to-date software and

databases.

Response: As suggested, we have used DADA2 in QIIME2 and a newer database SILVA to reanalyze our data sets and have updated the corresponding results. Compared with previous results, taxonomic composition changed as new pipeline and new database were used; however, the influence of updated abundance of genus on the efficiency of our tool was slight. In addition, as suggested by the reviewer, a more reliable network construction tool (SPIEC-EASI) was implemented in our method, which improves the efficiency of network integration (**Figure 2b-2c, Figure 2g**) and the prediction power of marker identification in our method (**Figure 4g**).

3. It would be useful to see examples and/or discussion on cases where the NetMoss method has shortcomings. This will help to understand the limitations of the method more deeply.

Response: We greatly appreciate the reviewer's comments to improve our study. The discussion on the limitation of NetMoss has been added in our updated manuscript (Page 14 Line 22-31).

4. Data and code availability. The analyses are based on simulations and publicly available data sets, all data is available. The source code is available through github but could be improved as follows:

- a) Add permanent DOI through e.g. Zenodo. This guarantees that the exact code version used in this manuscript will be preserved permanently.

See <https://guides.github.com/activities/citable-code/>

- b) In addition to the source code of the method, the repository should include code that was used to create the figures included in the manuscript. I did not find this information from the README or browsing the files quickly.

- c) The source code is missing license, hence it is not clear if the code is openly available (i.e. with an open license). Consider adding open source license on the code as is often recommended (see e.g. 10.1371/journal.pcbi.1002598)

- d) If the R scripts could be implemented as an R package, the method would be easier to use. The lack of this will remarkably limit the potential user base, and makes benchmarking with alternatives more difficult.

Consider including Rmarkdown vignette in the code that shows how to use the tools.

Response: As suggested, we have deposited our source code and datasets to public repositories.

- a) A permanent DOI has been added and the new version of our code could be accessed from Zenodo with DOI [10.5281/zenodo.5913042](https://doi.org/10.5281/zenodo.5913042)
- b) The code used to create the figures has been included in the previous version of our script. For the convenience of readers, we have improved it and have uploaded it to Zenodo <https://doi.org/10.5281/zenodo.5913042>
- c) License has been added in our source code.
- d) The source code of our method has been implemented as an R package NetMoss, which can be accessed from github <https://github.com/xiaolw95/NetMoss>
- e) Rmarkdown vignette has been included in our code.
See [in https://doi.org/10.5281/zenodo.5913042](https://doi.org/10.5281/zenodo.5913042)

Minor

The authors refer to "microbial interactions" but the taxonomic co-abundance networks are statistical associations rather than biological interactions, although in some cases these overlap. The abundance data itself does not, however, differentiate between statistical and biological interactions. I suggest to remove references to microbial interactions when discussing the methodology, and instead systematically use the term network module, co- occurrence, co-abundance patterns, or a similar term.

Response: As suggested, references to microbial interactions have been removed from the main text and we have replaced them with other terms (Page 15 Line 29,34,35; Page16 Line 1,7).

p5: tropological -> topological

Response: As suggested, the term has been revised (Page 5 Line 23).

SparCC has been used for taxonomic network detection. The SPIEC-EASI is a bit newer one and has shown considerable speed improvements with the same overall performance. It would be good to cite SPIEC-EASI in Discussion as an alternative method, and consider implementing that into the workflow.

Response: We thank the reviewer for this comment. In this revised manuscript, we have used SPIEC-EASI to replace SparCC to construct microbial networks. Moreover, we added a discussion of comparison among different network

construction methods (Page 12 Line 23-32). As shown in **Re. Fig. 1**, networks constructed by SPIEC-EASI were much sparser than those by the other two methods. Pearson-based method tended to build networks with more positive correlation edges (**Re. Fig. 1a-1b**). Although different tools lead to different network structure though, the markers identified from different networks largely overlapped (**Re. Fig. 1c**). Especially when using SPIEC-EASI, 84% of markers were shared by other tools (**Re. Fig. 1c**). The high overlap of SPIEC-EASI with other methods and great improvement of its application in network integration indicate the accordance with the real biological features, and thus this strategy was chosen to build network in our study.

Decision Letter, first revision:

Date: 14th March 22 23:53:16

Last Sent: 14th March 22 23:53:16

Triggered By: Jie Pan

From: jie.pan@us.nature.com

To: zhfq@biols.ac.cn

CC: computacionalscience@nature.com

Subject: AIP Decision on Manuscript NATCOMPUTSCI-21-0795B

Message: Our ref: NATCOMPUTSCI-21-0795B

14th March 2022

Dear Dr. Zhao,

Thank you for submitting your revised manuscript "Large-scale microbiome data integration enables robust biomarker identification" (NATCOMPUTSCI-21-0795B). It has now been seen by the original referees and their comments are below. The reviewers find that the paper has improved in revision, and therefore we'll be happy in principle to publish it in Nature Computational Science, pending minor revisions to satisfy the referees' final requests and to comply with our editorial and formatting guidelines. Please be kindly reminded that the remaining requests from referee #3 need to be satisfactorily addressed.

We are now performing detailed checks on your paper and will send you a checklist

detailing our editorial and formatting requirements in about a week. Please do not upload the final materials and make any revisions until you receive this additional information from us.

TRANSPARENT PEER REVIEW

Nature Computational Science offers a transparent peer review option for new original research manuscripts submitted from 17th February 2021. We encourage increased transparency in peer review by publishing the reviewer comments, author rebuttal letters and editorial decision letters if the authors agree. Such peer review material is made available as a supplementary peer review file. **Please state in the cover letter 'I wish to participate in transparent peer review' if you want to opt in, or 'I do not wish to participate in transparent peer review' if you don't.** Failure to state your preference will result in delays in accepting your manuscript for publication. Please note: we allow redactions to authors' rebuttal and reviewer comments in the interest of confidentiality. If you are concerned about the release of confidential data, please let us know specifically what information you would like to have removed. Please note that we cannot incorporate redactions for any other reasons. Reviewer names will be published in the peer review files if the reviewer signed the comments to authors, or if reviewers explicitly agree to release their name. For more information, please refer to our [FAQ page](https://www.nature.com/documents/nr-transparent-peer-review.pdf).

Thank you again for your interest in Nature Computational Science Please do not hesitate to contact me if you have any questions.

Sincerely,

Jie Pan, Ph.D.
Associate Editor
Nature Computational Science

ORCID

Reviewer #1 (Remarks to the Author):

The authors thoroughly revised the manuscript and addressed all the reviewers' questions and concerns. I am also glad to see that the suggested changes improved the final results. I do not have other concerns.

Reviewer #2 (Remarks to the Author):

All my raised concerns have been adequately addressed in the revised version. Therefore, the manuscript is suggested to be accepted.

Reviewer #3 (Remarks to the Author):

Thanks for responding adequately to most of my earlier comments.

My main remaining concern is that the new method should be more extensively and clearly compared with possible alternatives, to show its relative strengths as well as weaknesses, and to get a better idea about its overall application scope.

1. Benchmarking with alternatives has been added but remains somewhat limited in the results and figures. The response letter shows more comparisons ("Re. Fig. 3"), and according in many of these cases, the alternatives perform at the same level or better than NetMoss. These comparisons are not included in the main figures, instead there is just one Figure (4g) that shows a comparison where NetMoss outperforms all alternatives. The other benchmarks shown in "Re. Fig. 3" should also be included in the manuscript main figures; a table of AUC values might work better for comparison purposes than the current three distinct figures. Also more generally, additional benchmarking with alternative methods could help to establish the performance of the new method.
2. Why Fig 4c includes only NetMoss and Wilcoxon test, and not the Random Forest and PLS-DA that seemed to have good overall performance as well?
3. Figure 4f: is there a reason to assume that a larger number of biomarkers (from NetMoss) is a beneficial feature; could this be due to a larger number of false positives instead?
4. The main figures contain remarkable amount of material and elements. Splitting the information into multiple figures could help to more easily follow the message of the different figures. I find hard to follow the information in Fig. 5, for instance.
5. Figure 3e: what do the different ROC curves refer to? This is not mentioned in the figure or figure caption. Only average AUC is given across many curves, it would be informative to know how the individual curves (and their AUCs) compare between these alternative methods. This cannot be currently inferred from the figure.

Minor:

Fig. 1c color palette is not centered at 0 (white), should be I assume

Author Rebuttal, first revision:**Reviewer #1 (Remarks to the Author):**

The authors thoroughly revised the manuscript and addressed all the reviewers' questions and concerns. I am also glad to see that the suggested changes improved the final results. I do not have other concerns.

Response: We greatly appreciate the reviewer's comments on the novelty and significance of our study.

Reviewer #2 (Remarks to the Author):

All my raised concerns have been adequately addressed in the revised version. Therefore, the manuscript is suggested to be accepted.

Response: We thank very much for the reviewer's efforts on the improvement of our study.

Reviewer #3 (Remarks to the Author):

Thanks for responding adequately to most of my earlier comments.

My main remaining concern is that the new method should be more extensively and clearly compared with possible alternatives, to show its relative strengths as well as weaknesses, and to get a better idea about its overall application scope.

Response: We greatly appreciate the reviewer for these constructive comments.

1. Benchmarking with alternatives has been added but remains somewhat limited in the results and figures. The response letter shows more comparisons ("Re. Fig. 3"), and according in many of these cases, the alternatives perform at the same level or better than NetMoss. These comparisons are not included in the main figures, instead there is just one Figure (4g) that shows a comparison where NetMoss outperforms all alternatives. The other benchmarks shown in "Re. Fig. 3" should also be included in the manuscript main figures; a table of AUC values might work better for comparison purposes than the current three distinct figures. Also more generally, additional

benchmarking with alternative methods could help to establish the performance of the new method.

Response: Thank you. As suggested, we have added the detailed comparisons in **Figure 4f-i**. Considering that markers identified by NetMoss were based on network, which tended to capture more markers with subtle abundance changes, traditional abundance-based classification failed to utilize the advantage of NetMoss. For this reason, we updated the classification strategy of NetMoss by taking into account the importance of markers on the network. As shown in **Figure 4f**, NetMoss outperformed all other methods on the combined datasets (seven studies in total). Likewise, when applying these methods on each separate study, NetMoss was also better than the alternatives in 6 out of 7 studies (**Figure 4g-i**). Detailed descriptions have been added in the main text (Page 7 Line 236-239, Page 13 Line 421-424, Page 17 Line 573-576).

2. Why Fig 4c includes only NetMoss and Wilcoxon test, and not the Random Forest and PLS-DA that seemed to have good overall performance as well?

Response: We greatly appreciate the reviewer for this comment. As suggested, we have added a comparison including Random Forest and PLS-DA ("Fig 4c" in the comment may be a typo by the reviewer, which is actually not relevant to Wilcoxon test), demonstrating NetMoss's better performance. To avoid redundancy, we moved the result to **Suppl. Figure 6a** and added **Figure 4g-i** (see the response above).

3. Figure 4f: is there a reason to assume that a larger number of biomarkers (from NetMoss) is a beneficial feature; could this be due to a larger number of false positives instead?

Response: Thank you for pointing this out. Most of the CRC-related biomarkers identified by NetMoss have been confirmed by several previous studies, indicating that it may be a beneficial feature instead of introducing more false positives. The corresponding AUC values and the detailed bacterial taxa are shown in **Figure 4f** and **Suppl. Figure 5**.

4. The main figures contain remarkable amount of material and elements. Splitting the information into multiple figures could help to more easily follow the message of the different figures. I find hard to follow the information in Fig. 5, for instance.

Response: Thank you. Although each main figure in our manuscript is composed of multiple subfigures, they actually represent an integrated section and are addressing a

specific issue in the microbiome data integration. Splitting these figures would break the logic and flow of our manuscript. Nevertheless, we understand the reviewer's concern and to make our manuscript more favorable for the readers, we have added detailed descriptions of these figures in the main text (Page 6 Line 186-189, Page 7 Line 236-239, Page 12 Line 398-400, Page 13 Line 421-424, Page 17 Line 573-576).

5. Figure 3e: what do the different ROC curves refer to? This is not mentioned in the figure or figure caption. Only average AUC is given across many curves, it would be informative to know how the individual curves (and their AUCs) compare between these alternative methods. This cannot be currently inferred from the figure.

Response: Sorry for the unclear description about the ROC curves. The different ROC curves in Figure 3e refer to the prediction power of ten times of replication under different noise levels, which indicate that NetMoss shows a stable and superior performance over previous methods. Details also can be seen in **Suppl. Figure 4**. We have clarified this in the main text (Page 6 Line 186-189, Page 12 Line 398-400) and listed the corresponding AUC values in **Suppl. Table 2**.

Minor:

Fig. 1c color palette is not centered at 0 (white), should be I assume

Response: Thank you for pointing it out. The color palette has been updated in this new version.

Final Decision Letter:

Date: 12th April 22 09:41:24

Last Sent: 12th April 22 09:41:24

Triggered By: Jie Pan

From: jie.pan@us.nature.com

To: zhfq@biols.ac.cn

Subject: Decision on Nature Computational Science manuscript NATCOMPUTSCI-21-0795C

Message: Dear Professor Zhao,

We are pleased to inform you that your Article "Large-scale microbiome data integration enables robust biomarker identification" has now been accepted for publication in Nature Computational Science.

Please note that *Nature Computational Science* is a Transformative Journal (TJ). Authors may publish their research with us through the traditional subscription access route or make their paper immediately open access through payment of an article-processing charge (APC). Authors will not be required to make a final decision about access to their article until it has been accepted. [Find out more about Transformative Journals](https://www.springernature.com/gp/open-research/transformative-journals)

Authors may need to take specific actions to achieve [compliance with funder and institutional open access mandates](https://www.springernature.com/gp/open-research/funding/policy-compliance-faqs). If your research is supported by a funder that requires immediate open access (e.g. according to [Plan S principles](https://www.springernature.com/gp/open-research/plan-s-compliance)) then you should select the gold OA route, and we will direct you to the compliant route where possible. For authors selecting the subscription publication route, the journal's standard licensing terms will need to be accepted, including [self-archiving policies](https://www.springernature.com/gp/open-research/policies/journal-policies). Those licensing terms will supersede any other terms that the author or any third party may assert apply to any version of the manuscript.

Acceptance of your manuscript is conditional on all authors' agreement with our publication policies (see <https://www.nature.com/natcomputsci/for-authors>). In particular your manuscript must not be published elsewhere and there must be no announcement of the work to any media outlet until the publication date (the day on

which it is uploaded onto our web site).

Before your manuscript is typeset, we will edit the text to ensure it is intelligible to our wide readership and conforms to house style. We look particularly carefully at the titles of all papers to ensure that they are relatively brief and understandable.

Once your manuscript is typeset and you have completed the appropriate grant of rights, you will receive a link to your electronic proof via email with a request to make any corrections within 48 hours. If, when you receive your proof, you cannot meet this deadline, please inform us at rjsproduction@springernature.com immediately.

If you have queries at any point during the production process then please contact the production team at rjsproduction@springernature.com. Once your paper has been scheduled for online publication, the Nature press office will be in touch to confirm the details.

Content is published online weekly on Mondays and Thursdays, and the embargo is set at 16:00 London time (GMT)/11:00 am US Eastern time (EST) on the day of publication. If you need to know the exact publication date or when the news embargo will be lifted, please contact our press office after you have submitted your proof corrections. Now is the time to inform your Public Relations or Press Office about your paper, as they might be interested in promoting its publication. This will allow them time to prepare an accurate and satisfactory press release. Include your manuscript tracking number NATCOMPUTSCI-21-0795C and the name of the journal, which they will need when they contact our office.

About one week before your paper is published online, we shall be distributing a press release to news organizations worldwide, which may include details of your work. We are happy for your institution or funding agency to prepare its own press release, but it must mention the embargo date and Nature Computational Science. Our Press Office will contact you closer to the time of publication, but if you or your Press Office have any inquiries in the meantime, please contact press@nature.com.

We welcome the submission of potential cover material (including a short caption of around 40 words) related to your manuscript; suggestions should be sent to Nature Computational Science as electronic files (the image should be 300 dpi at 210 x 297 mm in either TIFF or JPEG format). We also welcome suggestions for the Hero Image, which appears at the top of our [home page](http://www.nature.com/natcomputsci); these should be 72 dpi at 1400 x 400 pixels in JPEG format. Please note that such pictures should be selected more for their aesthetic appeal than for their scientific content, and that colour images work better than black and white or grayscale images. Please do not try to design a cover with the Nature Computational Science logo etc., and please do not submit composites of images related to your work. I am sure you will understand that we cannot make any promise as to whether any of your suggestions might be selected

for the cover of the journal.

Best regards,

Jie Pan, Ph.D.
Associate Editor
Nature Computational Science

P.S. Click on the following link if you would like to recommend Nature Computational Science to your librarian: https://www.springernature.com/gp/librarians/recommend-to-your-library

** Visit the Springer Nature Editorial and Publishing website at www.springernature.com/editorial-and-publishing-jobs for more information about our career opportunities. If you have any questions please click here. **